# Analgesic Effect of Intrathecal Morphine Combined with Low-Dose Bupivacaine on Postoperative Analgesia after Liver Resection: A Randomized Controlled Study

**DOI:** 10.3390/jpm12020211

**Published:** 2022-02-03

**Authors:** MinGi Ban, Yong Seon Choi, Bon-Nyeo Koo

**Affiliations:** Department of Anesthesiology and Pain Medicine, Anesthesia and Pain Research Institute, Yonsei University College of Medicine, Seoul 03722, Korea; banmj09@yuhs.ac (M.B.); yschoi@yuhs.ac (Y.S.C.)

**Keywords:** intrathecal, bupivacaine, analgesia

## Abstract

Although intrathecal morphine and bupivacaine are increasingly implemented in effective postoperative pain control, there is a lack of consensus on the dosage as high doses of bupivacaine may inadvertently cause unwanted side effects. The purpose of this study was to compare the effects of intrathecal morphine injection and low-dose bupivacaine with morphine injection. In total, 90 patients were divided into 3 groups: (1) sham injection for the control group; (2) morphine 400 mcg for the morphine group (M); and (3) morphine 400 mcg and bupivacaine 5 mg for the morphine and bupivacaine group (M + B). Our primary outcome was time to first rescue analgesic. The VAS (visual analogue scale) pain score was compared until POD (postoperative day)1. Total fentanyl dose was compared until POD2. Side effects were monitored until POD3. Although time to first rescue was significantly shorter in the control group compared to group M and group M + B (*p* < 0.001), both groups (M and M + B) were comparable to each other. There was a significant decrease in the VAS score and total fentanyl administration in group M and group M + B compared to the control group. Pruritus and tingling were more prevalent in the M + B group (*p* = 0.023; *p* = 0.010). The addition of 5 mg bupivacaine may be insufficient in providing further analgesic benefits; however, higher doses may aggravate side effects.

## 1. Introduction

Effective postoperative pain control plays a vital part in the management of patients after major surgeries, including hepatectomy. A proper postoperative pain control plan will promote patient satisfaction while decreasing time to ambulation, respiratory and cardiovascular complications, and consequently mortality [1,2]. Recently, a multimodal approach rather than a pain control method relying on opioids alone has been advocated for to control postoperative pain more effectively while reducing the side effects of drugs [3]. The Enhanced Recovery After Surgery (ERAS) guidelines emphasize the significance of multimodal analgesia, and the goal of optimizing postoperative analgesia while avoiding potential side effects is increasingly becoming a primary interest after major operations, including liver resection [4].

One method of the multimodal approach includes postoperative epidural patient-controlled analgesia (PCA). However, for liver resection patients, due to the possible complications from coagulopathies, cost-effectiveness of the procedure, and excessive sympathetic block by epidural analgesia, continuous placement of an epidural catheter has been debated against [5,6,7,8]. Continuous placement of an epidural catheter after liver resection is a major concern because of postoperation coagulopathy and increased risks of spinal hematoma [4,6]. Intrathecal morphine with the combined use of an intravenous (IV) PCA is a method that avoids the potential complications of an epidural catheter and studies have shown that the analgesic effects are not reduced compared to an epidural [8,9,10]. Guidelines for liver surgery published by ERAS strongly recommend intrathecal opioids instead of epidural analgesia as part of multi-modal analgesia [11,12].

Intrathecal morphine used as a ‘one-shot’ method provides several advantages as it is easy, cost-effective, and reliable, and technical failures are rare [13]. The analgesic effects last for 20–48 h [14,15]. However, because of the hydrophilic properties of morphine, the peak effect time of an intrathecal injection is 6 h, which means morphine alone may not be adequate for immediate postoperative pain control [10]. To compensate for this time, some studies have suggested that the use of a combination of local anesthetic, such as bupivacaine, is effective during this period. However, the possible complications and optimal dose have not been evaluated sufficiently for liver resection. Although previous studies have suggested a high dose of bupivacaine could induce undesired excessive motor block and hemodynamic changes during surgery, there is a lack of evidence of the effects of lower doses of bupivacaine [16]. Koning et al. suggested that the addition of 12.5 mg of bupivacaine to morphine during robot-assisted radical prostatectomy reduced opioid consumption and was a viable multimodal analgesic postoperative method [14]. A lower dose of 5 mg combined with morphine was compared against saline and shown to be effective in lowering pain scores and opioid consumption without adverse effects. Although bupivacaine may induce unwanted side effects after surgery, including motor block, sensory block, and tingling sensation, no studies have investigated whether low-dose bupivacaine would have synergistic or additional analgesic effects to intrathecal morphine while reducing any risk of side effects with respect to postoperative pain control.

The aim of our study was to compare the effectiveness and side effects of intrathecal morphine combined with low-dose bupivacaine against intrathecal morphine alone and no intrathecal injection. We hypothesized that the addition of low-dose bupivacaine would provide improved immediate postoperative analgesia and delay the time to first rescue analgesics. Moreover, we compared opioid consumption and pain scores during the initial 48 h, and side effects in the first 72 h postoperative.

## 2. Materials and Methods

After Institutional Review Board (IRB no. 4–2018–0838) approval, we conducted a single-center double-blinded randomized prospective clinical trial in a teaching hospital from October 2018 to April 2020. The study protocol was registered at www.clinicaltrials.gov (NCT05208801, 26 January 2022). Patients over the age of 19 years scheduled for liver resection under open or laparoscopic surgery were eligible for participation. Exclusion criteria included: contraindication to spinal anesthesia (including coagulation disorders, increased intracranial pressure, severe systemic infection); contraindication to study medication (including allergies); patients with psychological or neurological disorders that affect pain assessment; patients with severe respiratory, heart, or kidney disease; and patients unable to read the consent form (including illiteracy and mental disorders).

Patients were informed about the purpose and method of the study. After explanation, the patient was revisited at least 1 day after the initial explanation and written consent was obtained. Explanation of the purpose of the study and written consent was conducted in an independent counseling office on the ward

On the day of surgery, the study subjects were randomly divided into 3 groups using a random number table: (1) control group; (2) morphine administration group (group M); and (3) or morphine + bupivacaine administration group (group M + B). The patient, surgical team, nurses on the ward, and researchers were blinded to the allocation. Patient group assignment, drug preparation, and drug administration were provided by a trained anesthesiologist uninvolved in the rest of the study. To ensure blinding, the attending anesthesiologist was excluded from the operation room until the spinal procedure and initial dermatomal assessment were complete. Thereafter, the attending anesthesiologist who was blinded to group allocation initiated general anesthesia.

### 2.1. Intrathecal Injection

Upon arrival in the operating room, standard monitoring, including an electrocardiogram, non-invasive blood pressure monitor, and pulse saturation, was initiated after confirming the patient. All patients received an intrathecal injection prior to general anesthesia. Patients were placed in a lateral decubitus position and after performing skin sterilization, a trained anesthesiologist infiltrated the skin with 1% lidocaine using a 25G needle for local anesthesia.

In the control group, a sham procedure of 2 mL of 1% lidocaine injected percutaneously using the initial 25G needle used for local anesthesia was performed. For group M and group M + B, after confirming proper placement of the needle in the spinal canal by cerebrospinal fluid using a 25G pencil-point spinal needle, 400 mcg of morphine or 400 mcg of morphine with 5 mg of 0.5% bupivacaine chloride were injected, respectively. The patient was positioned back in the supine position and after 5 min, we checked the spread of dermatome to the spinal injection using an alcohol swab to test at which point the patient felt sensory loss.

### 2.2. Anesthesia Method

For all groups, standardized general anesthesia typical of liver resection was administered after spinal puncture. Pre-medication with glycopyrrolate 0.1 mg IV was administered before induction of anesthesia. Induction of anesthesia was performed with propofol and remifentanil and injection of 0.6 mg/kg rocuronium for sufficient muscle relaxation and tracheal intubation. Anesthesia was maintained with total intravenous anesthesia, propofol sustained with an infusion of propofol-target controlled infusion (TCI) using the target concentration control injector Marsh model, and remifentanil maintained in a remifentanil-TCI continuous infusion using the Minto model. For hemodynamic monitoring during surgery, arterial cannulation and intravenous cannulation were performed.

During surgery, blood pressure, central venous pressure, and cardiac output were continuously monitored. In cases of hypotension (blood pressure or heart rate within 20% of the baseline), blood pressure was controlled by adjusting the anesthetic concentration, fluid supply, blood transfusion or inotropes, or vasopressor depending on the cause. In addition, a bispectral index (BIS) monitor capable of checking the depth of anesthesia during surgery was applied and anesthesia depth was maintained between 40 and 60. A FloTrac/Vigileo System (Edwards Lifescience LLC, Irvine, CA, USA) was used to monitor the cardiac index, stroke volume variation (SVV), systemic vascular resistance index (SVRI), and mean arterial pressure (MAP). We managed fluid therapy using a goal-directed strategy (SVV < 13%, MAP > 75 mmHg, CI ≥ 2.0 L/min/m^2^, SVRI ≤ 3000 dynes s cm^−5^/m^2^). All procedures are standardized in our institute.

After closure of the peritoneum, fentanyl 1 mcg × body weight (BW) and nefopam hydrochloride 40 mg (Acupan, Korea) were administered intravenously for postoperative pain control, and 0.3 mg ramosetron for the prevention of nausea and vomiting.

At the end of surgery, propofol and remifentanil infusions were ceased, and neostigmine and glycopyrrolate were used to reverse muscle relaxation. When the patient’s consciousness and muscle relaxation was restored, we extubated the patient and transferred the patient to the postoperative anesthesia care unit (PACU). All patients received an intravenous patient-controlled analgesia (IV PCA) regimen, which was started at the end of surgery, as follows: fentanyl 15 × BW (mcg) with ramosetron 0.6 mg and normal saline to a total volume of 100 mL. The infusion rate was set at 1 mL per hour with a bolus of 1 mL and a lockout time of 7 min.

### 2.3. In the PACU

Standard vital signs, including 3-lead ECG, blood pressure, and oxygen saturation, were monitored in the PACU. At 30 and 60 min after entering the PACU, an anesthesiologist blind to the patients’ assignment assessed whether the patient presented any side effects of the intrathecal injection by assessing the patient’s sensory and motor block levels and whether the patient had a tingling sensation. Patients presenting with sensory and motor block levels were observed in the PACU until the sensory or motor block dissipated. Moreover, any possible adverse reactions to intrathecal injection, including headache, nausea, vomiting, pruritus, shivering, respiratory depression, decreased consciousness, and hypotension, were recorded. At any time during this study, respiratory depression was defined as less than or equal to 8 breaths/min. Postoperative sedation using the 8-point modified Ramsey Sedation Scale (RSS) was used to evaluate postoperative sedation and over-sedation was defined as RSS greater than 4. Patients were excluded from the study if over-sedation persisted for more than 1 h. Hypotension was defined as a 15% decrease in systolic blood pressure from baseline.

The anesthesiologist also assessed the patient for pain using the visual analogue scale (VAS) in the PACU at 30 and 60 min to compare immediate postoperative pain control amongst the groups. If the patient presented with a pain score of VAS 5 or higher, fentanyl 1 μg/kg IV was administered.

### 2.4. In the Ward

In the ward, a researcher, blind to the patient’s assignment group, checked the patient for possible side effects, including sensory and motor block level, tingling sensation, headache, nausea, vomiting, pruritus, respiratory depression, decreased consciousness, and hypotension, on the night of surgery and on the day after surgery. To compare postoperative pain control among the groups, the pain score was assessed using VAS on the night of surgery and the day after surgery. For further assessment of pain control, the number of painkillers administered on postoperative day (POD) 1, 2, and 3 and the total dose of fentanyl administered through IV PCA was recorded. Moreover, we checked the time to ambulation for each patient.

### 2.5. Rescue Analgesics

If the patient complained of a pain score of 5 or higher on the VAS scoring system, despite the use of IV-PCA in the ward, rescue analgesic (intravenous pethidine 25 mg) was given. To compare the effectiveness of the analgesia, we compared whether the patients received a first rescue analgesic and if they did, the time to the first rescue analgesic. Moreover, subsequent rescue analgesics administered on POD 1, 2, and 3 were recorded. Total analgesic consumption was recorded in the first 24 h postoperation and on the following POD 2 and 3.

### 2.6. Power of Study

The primary end point was the time to first rescue analgesic in the first 72 h.

To detect a difference of one SD between the mean time to first rescue analgesic (1), a sample size of 28 patients for each group was required to have a power of 80% with α = 0.025 (one-sided hypothesis). Taking into consideration the potential for drop-outs, we decided to enroll 30 patients per group.

Data are expressed as mean ± SD of the mean for continuous values or median with interquartile range for discontinuous values. The time to first rescue was described as a median and interquartile range (IQR) and values were compared using the Kaplan–Meier statistic. Distributions were examined to ensure proper statistical treatment. Data were analyzed for normal distribution and one-way ANOVA with Bonferroni correction for multiple comparisons was performed for continuous data. For ordinal data, the Chi-Square Test was used. A *p*-value < 0.05 was deemed statistically significant and a *p*-value < 0.01 was deemed statistically significant for secondary outcomes after correction. Values were calculated with Statistical Package for Social Scienced statistical software (version 23.0, SPSS An IBM Company, Chicago, IL, USA).

## 3. Results

A total of 91 patients were screened for enrollment, of whom 1 patient was not included due to the cancellation of surgery as shown in Figure 1. Four patients withdrew consent after random allocation, during the period of the study.

A total of 86 patients were analyzed, with 28 patients in the control group, 28 patients in group M, and 30 patients in group M + B. Demographics and surgical characteristics were balanced at baseline and are shown in Table 1.

Extubation time was defined from the moment anesthetics were ceased until the patient was extubated.

The numbers are the mean ± SD or number of patients (percentage, %).

The median (IQR) time to first rescue, our primary endpoint, was 13 (8–18) min for the control group, which was significantly shorter compared to group M and group M + B (60 (34–86) min vs. 70 (21–118) min, *p* < 0.001); however, there was no significant difference between group M and M + B, as shown in Figure 2.

There was a significant reduction in the mean VAS score in group M + B and group M compared to the control group until POD 1; however, there was no significant difference in VAS scores at any time point between group M and group M + B (Table 2). Patients in the control group required more rescue analgesics compared to both the M group and the M + B group on POD 1 (*p* < 0.001). There was no difference in rescue analgesics between group M and M + B group. (Table 2). On POD 2 and 3, there were no significant differences among groups (*p* = 0.0702, *p* = 0.159, respectively) (Table 2).

Patients in the control group required more rescue analgesics compared to both the M group and the M + B group on POD 1 (*p* < 0.001). There was no difference in rescue analgesics between group M and M + B group (Table 2). On POD 2 and 3, there were no significant differences among groups (*p* = 0.0702, *p* = 0.159, respectively) (Table 2). The fentanyl dose administered by PCA was significantly lower in group M and group M + B compared to the control group from the surgery night (*p* < 0.001) until POD 1 (*p* = 0.006), with no difference on POD 2 (*p* = 0.488). There was no difference between the fentanyl dose between group M and group M + B throughout the study period until POD2 (Table 2).

In total, 12 patients in the M + B group (40%) presented with sensory block compared to 0 patients in the control group and 2 patients in the M group (7.1%) (*p* < 0.001). In total, 4 patients in the M + B group (13.3%) showed motor block compared to 0 patients in the control group and 1 patient in the M group (*p* = 0.079). All cases of sensory and motor block were resolved before leaving the PACU. Side effects are shown in Table 3. All 6 patients that experienced tingling in the first 72 h were in the M + B group (*p* = 0.010). Out of the 14 patients that experienced pruritus, 1 patient was in the control group, 4 patients in the M group, and 9 patients in the M + B group, showing a significant increase in the M + B group compared to the control group (Table 3). There was no significant difference in the other side effects, including headache, postoperative nausea and vomiting (PONV), respiratory depression, somnolence, shivering, and hypotension (*p* > 0.05) (Table 3). Time to ambulation was comparable in all three groups (23.7 ± 5.3 h in the control group vs. 23.3 ± 6.3 h in the M group vs. 25.1 ± 8.6 h in the M + B group) (*p* = 0.267).

Propofol and remifentanil requirements during surgery were comparable in the three groups (*p* = 0.825, *p* = 0.772, respectively). The intraoperative hemodynamic parameters, as shown in Table 4, show that there were no significant differences in HR, MBP, CVP, or BIS among the three groups during the operation. FloTrac indices also showed no significant differences in CI, SVV, or SVRI among the 3 groups (*p* > 0.05) (Table 4). Postoperative MBP showed no difference at 30 min upon arrival at the PACU (*p* = 0.129) or at 60 min upon arrival at the PACU (*p* = 0.336).

## 4. Discussion

In this study, the addition of low-dose bupivacaine to intrathecal morphine failed to show supplementary analgesics benefits in comparison to intrathecal morphine injected singularly. We revealed that there were analgesic benefits of intrathecal morphine injection and intrathecal morphine with bupivacaine injection in comparison to the control group, but neither group was superior to the other after liver resection. The patients reported significantly lower pain scores and overall opioid consumption was decreased via IV PCA in both intervention groups in comparison to the control group. Immediate postoperative pain control reflected by time to first rescue analgesic was not significantly superior in the bupivacaine with morphine group compared to the morphine group.

A relatively low dose of morphine (e.g., <500 mcg) with a local anesthetic regimen is suggested to provide optimal analgesic benefits with decreased side effects in patients. However, a consensus on bupivacaine dosage is lacking. Lemoine et al. suggested that the optimal spinal dose of bupivacaine for the recovery of motor function and guaranteed hospital discharge in patients undergoing ambulatory surgery was 7.5 mg as this dose resolved motor block within 5 h and achieved discharge within 6 h in 95% of patients [17]. However, Karamuz et al. demonstrated that 7.5 mg intrathecal bupivacaine resulted in higher incidences of side effects, including hypotension and shivering, compared to bupivacaine 4 mg combined with fentanyl 25 mcg, which provided adequate anesthesia for transurethral prostatectomy [18]. Guidaityte et al. reported that intrathecal injection of 4 mg and 5 mg intrathecal bupivacaine provided sufficient anesthesia for anorectal surgery with a sensory block duration of 4 to 5 h, with a maximum VAS at 6 h [19]. A higher dose of 7.5 mg intrathecal bupivacaine provided a longer duration of both sensory and motor block. Motamed et al. showed that 5 mg bupivacaine with morphine (75 or 100 mcg) was effective in postoperative analgesia for elective laparoscopic cholecystectomy [20]. To our knowledge, no previous studies have been performed that have investigated low-dose bupivacaine with morphine against morphine in liver resection. To avoid further induction of side effects brought by a higher dose, 5 mg was used in this study. Because an ideal analgesic method has maximal benefits with the lowest possible side effects, verifying the synergic effects of low-dose bupivacaine with intrathecal morphine could enhance recovery from major abdominal surgeries, including liver resection.

Unfortunately, our study suggested that morphine injected intrathecally alone at a low dose of 400 mcg is comparable to morphine combined with 5 mg of bupivacaine, and there were no additional benefits.

Koning et al. investigated the use of 12.5 mg bupivacaine with 300 mcg morphine intrathecally in 150 patients that underwent robot-assisted radical prostatectomy [14]. They found a mean reduction in IV opioids during admission and lower pain scores, which was reflected in our study as well. An increase in the bupivacaine dosage may provide additional analgesic effects; however, a higher dose of intrathecal bupivacaine may also induce unwanted hemodynamic disturbances and undesired side effects, such as tingling sensation and sensory and motor block. Increasing the dosage of bupivacaine may warrant a further decrease in perioperative blood pressure, causing episodes of hypotension that require intervention. Although previous studies investigated a higher dosage of bupivacaine as an additive without further unwanted hemodynamic side effects, different factors may aggravate hemodynamic disturbances, such as age, hypovolemia, and possibly differences in race. Especially in the case of hepatectomy, intraoperative restrictive fluid management is often required, which predisposes a higher risk for hypotension and a deeper sympathetic block for postoperative pain management may aggravate this risk [21]. When increasing the dose of bupivacaine for liver resection, this should be taken into consideration, carefully weighing the risks and benefits. It is necessary to meticulously adjust the dosage of additive bupivacaine and further studies are warranted to evaluate the optimal dosage of local anesthetic additive to intrathecal morphine.

Pruritus was increased in both intervention groups, which was in accordance with other studies. Although bupivacaine has been reported to reduce the incidence of opioid-induced pruritus by interfering with local neuronal blockade or mu opioid receptors, our results do not reflect this [22]. Other factors that would cause pruritus were not evaluated in this study. Prophylactic drugs against pruritus, including ondansetron and dehydrobenzperidol, were not administered in this study. The inclusion of a prophylactic measure and continuation of 5-HT3 antagonists may have decreased the incidence of pruritus in the intervention groups.

Respiratory depression was not present in this study. Studies that report late-onset respiratory depression that required intervention due to intrathecal morphine usually presented cases with higher doses of morphine (>500 mcg) [23]. Clinically relevant respiratory depression has been shown to be unlikely to occur with lower doses of intrathecal morphine and thereby, we did not institute specific monitoring for respiratory depression overnight.

Although there are a variety of possible analgesic methods, including peripheral nerve blocks for hepatectomy, for achieving multimodal analgesia, intrathecal injection is a relatively easier method to perform in the clinical field. Thereby, it is important to assess the additive or synergic analgesic effects of bupivacaine to find the optimal dose. However, because it has been suggested that a higher dose may bring further unwanted side effects, further studies should take this into consideration and carefully weigh the risk and benefits. Other intrathecal regimens should also be taken into consideration for postoperative pain control after liver resection. Alpha 2 adrenoreceptor antagonists, including clonidine and dexmedetomidine, are increasingly being acknowledged as a local anesthetic adjuvant. Crespo et al. suggested that intrathecal clonidine is a safe adjuvant to neuraxial anesthesia in prolonging sensory block and motor block without increasing hypotension, pruritus, or PONV. Other drugs, including ketamine and steroids, have been used with mixed results [24]. However, these regimens require further research in regards to the safety profile.

This study has limitations. First, because this study was performed in a single-center institution with a small sample size, there is a disadvantage of a lack of generalizability. Secondly, this study evaluated a single dose of bupivacaine. Further studies may be needed to identify the optimal regimen for hepatectomy.

## 5. Conclusions

Intrathecal morphine alone can be effectively implemented in a multimodal analgesic approach to reduce overall opioid consumption during the first 24 h of postoperative care after liver resection. The addition of 5 mg of bupivacaine was insufficient in providing analgesic benefits in combination with intrathecal morphine but resulted in sensory and motor block and tingling sensation in the case of hepatectomy. Therefore, intrathecal analgesia may warrant additional study to identify the optimal regimen.

## Figures and Tables

**Figure 1 jpm-12-00211-f001:**
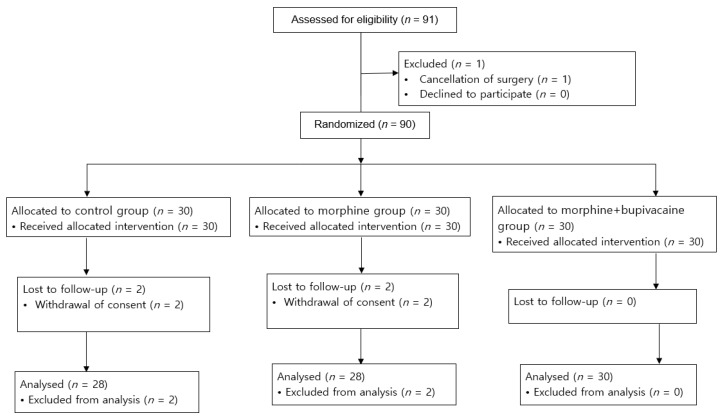
Flow diagram of the study selection process.

**Figure 2 jpm-12-00211-f002:**
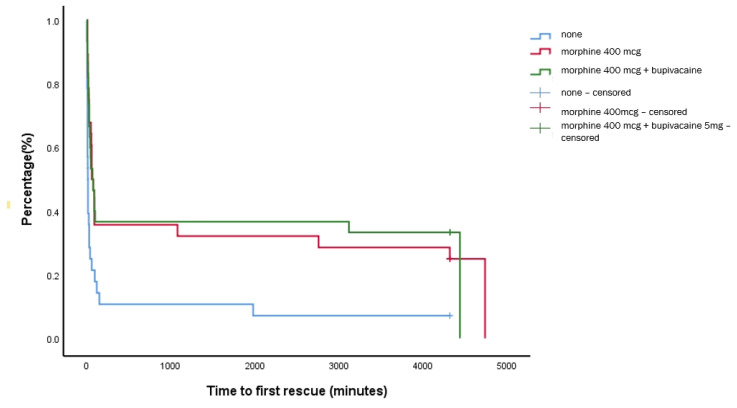
Kaplan–Meier analysis of the time to first rescue analgesic.

**Table 1 jpm-12-00211-t001:** Demographic data and intraoperational characteristics.

	Group Control (*n* = 28)	Group M (*n* = 28)	Group M + B (*n* = 30)	*p*
Sex				0.629
Male	21 (57.0%)	19 (67.9%)	19 (63.3%)	
Age	44.9 ± 17.9	43.0 ± 14.4	38.2 ± 13.1	0.217
Body weight (kg)	71.1 ± 10.8	66.6 ± 10.2	66.8 ± 11.2	0.224
Height (cm)	170.8 ± 8.3	167.2 ± 9.8	166.9 ± 9.0	0.199
BMI (kg/m^2^)	24.3 ± 2.8	23.5 ± 1.9	23.9 ± 2.8	0.487
Anesthesia time (min)	346.1 ± 135.5	410.4 ± 102.5	411.8 ± 103.4	0.053
Operation time (min)	280.3 ± 131.5	336.4 ± 100.9	338.3 ± 99.4	0.089
Operation type				0.752
Open	16 (57.1%)	17 (60.7%)	20 (66.7%)	
Laparoscopy	12 (42.9%)	11 (39.3%)	10 (33.3%)	
Extent of resection				0.557
Right lobe	15 (53.6%)	19 (67.9%)	23 (76.7%)	
Left lobe	4 (14.3%)	4 (14.3%)	2 (6.7%)	
Central	1 (3.6%)	1 (3.6%)	0 (0.0%)	
Segment	8 (28.6%)	4 (14.3%)	5 (16.7%)	
Extubation time (min)	15.9 ± 12.9	13.6 ± 6.3	14.7 ± 4.8	0.634

BMI: body mass index.

**Table 2 jpm-12-00211-t002:** Pain score and additional analgesics required in the postoperative period.

	Group Control (*n* = 28)	Group M (*n* = 28)	Group M + B (*n* = 30)	*p*
VAS in PACU (30 min)	6.1 ± 2.6	3.7 ± 1.9 *	4.1 ± 2.9 *	0.001
VAS in PACU (60 min)	5.5 ± 2.4	3.4 ± 1.9 *	3.3 ± 2.4 *	<0.001
VAS on operation night	4.4 ± 1.7	2.6 ± 1.3 *	2.4 ± 1.2 *	<0.001
VAS POD1	3.5 ± 1.5	2.4 ± 1.7 *	2.2 ± 1.1 *	0.002
IV PCA Fentanyl Dose (mcg) Op night	227.3 ± 117.9	122.1 ± 58.8 *	136.6 ± 63.3 *	<0.001
IV PCA Fentanyl Dose (mcg) POD 1	365.7 ± 218.0	214.0 ± 166.7 *	253.6 ± 142.9 *	0.006
IV PCA Fentanyl Dose (mcg) POD 2	176.3 ± 147.6	210.0 ± 159.8	226.3 ± 172.3	0.488
Additional analgesics POD 1	1.5 ± 1.4	0.3 ± 0.6 *	0.3 ± 0.6 *	<0.001
Additional analgesics POD 2	0.5 ± 0.8	0.5 ± 0.7	0.3 ± 0.8	0.702
Additional analgesics POD 3	0.3 ± 0.5	0.5 ± 1.1	0.2 ± 0.4	0.159

VAS: visual analogue scale; PACU: postoperative anesthesia care unit; POD: postoperative operative day; IV: intravenous; PCA: patient-controlled analgesia. The numbers are the mean ± SD or number of patients (percentage, %). * *p* < 0.001 vs. group control.

**Table 3 jpm-12-00211-t003:** Side effects of intrathecal injection for postoperative 3 days.

	Group Control	Group M	Group M + B	*p*
Headache	3 (10.7%)	2 (7.1%)	6 (20.0%)	0.316
PONV	8 (28.6%)	9 (32.1%)	11 (36.7%)	0.804
Pruritus	1 (3.6%)	4 (14.3%)	9 (30.0%) *	0.023
Respiratory depression	0 (0%)	0 (0%)	0 (0.0%)	
Somnolence	16 (57.1%)	18 (64.3%)	19 (63.3%)	0.836
Hypotension	1 (3.6%)	2 (7.1%)	3 (10.0%)	0.630
Tingling	0 (0.0%) ^†^	0 (0.0%) ^†^	6 (20.0%)	0.010
Shivering	3 (10.7%)	4 (14.3%)	2 (6.7%)	0.638

PONV: postoperative nausea and vomiting. The numbers are the number of patients (percentage, %). * *p* < 0.01 vs. group control. ^†^
*p* < 0.05 vs. group M + B.

**Table 4 jpm-12-00211-t004:** Intraoperative findings.

	Group Control (*n* = 28)	Group M (*n* = 28)	Group M + B (*n* = 30)	*p*
HR (bpm)				
Initial	72.3 ± 14.3	74.6 ± 15.7	74.1 ± 13.2	0.811
Induction	69.7 ± 11.3	71.4 ± 14.2	69.6 ± 12.9	0.845
Skin incision	60.9 ± 9.2	62.4 ± 11.9	61.9 ± 10.9	0.856
2 h	69.5 ± 13.1	68.2 ± 11.9	69.8 ± 11.2	0.865
3 h	68.2 ± 9.9	66.9 ± 11.7	69.6 ± 10.4	0.645
4 h	72.2 ± 11.4	70.5 ± 12.9	73.6 ± 9.8	0.669
End of surgery	78.2 ± 19.0	71.8 ± 15.1	78.0 ± 15.2	0.249
MAP (mmHg)				
Initial	91.9 ± 15.1	92.7 ± 11.3	88.7 ± 14.7	0.506
Induction	77.4 ± 13.5	77.0 ± 10.0	76.9 ± 15.0	0.990
Skin incision	77.6 ± 13.0	75.5 ± 10.8	75.5 ± 12.3	0.756
2 h	88.6 ± 10.2	86.0 ± 9.9	88.6 ± 10.4	0.545
3 h	83.4 ± 8.8	83.4 ± 8.6	84.6 ± 11.2	0.884
4 h	85.8 ± 10.3	79.7 ± 11.1	85.1 ± 10.7	0.144
End of surgery	93.2 ± 14.8	85.5 ± 11.5	91.6 ± 15.3	0.102
CVP (cmH_2_0)				
Initial	-			
Induction	5.8 ± 2.4	5.8 ± 4.0	5.9 ± 2.4	0.998
Skin incision	6.0 ± 2.5	5.7 ± 5.7	5.5 ± 2.4	0.728
2 h	5.2 ± 2.0	4.9 ± 2.3	4.1 ± 2.9	0.272
3 h	4.7 ± 1.7	4.6 ± 2.1	4.8 ± 1.7	0.926
4 h	5.0 ± 1.5	4.5 ± 2.4	4.9 ± 2.0	0.759
End of surgery	6.0 ± 2.3	6.0 ± 2.8	5.9 ± 2.3	0.978
BIS				
Initial	99.2 ± 2.7	98.7 ± 2.9	98.9 ± 2.0	0.785
Induction	33.9 ± 9.0	41.3 ± 12.3	39.4 ± 12.9	0.052
Skin incision	30.1 ± 8.7	31.9 ± 7.2	32.5 ± 7.8	0.491
2 h	29.4 ± 7.3	30.2 ± 6.0	31.6 ± 7.0	0.470
3 h	30.7 ± 6.9	33.4 ± 5.4	32 ± 5.4	0.257
4 h	34.4 ± 6.0	34.2 ± 6.9	34.8 ± 5.3	0.955
End of surgery	41.8 ± 12.0	40.5 ± 10.7	38.5 ± 9.0	0.491
GDFT ^1^_CI ≥ 2.0 (%)	78.9 ± 29.0	86.3 ± 20.8	86.5 ± 17.9	0.464
GDFT_SVRI ≤ 3000 (%)	95.8 ± 11.9	97.3 ± 7.7	99.5 ± 1.9	0.113
GDFT_MAP ≥ 75 (%)	86.8 ± 14.0	76.0 ± 21.9	78.9 ± 19.6	0.055
GDFT_SVV ≤ 13 (%)	87.9 ± 17.6	76.7 ± 27.4	82.3 ± 24.7	0.190

^1^ Goal-directed fluid therapy was practiced in our study to maintain proper fluid management for our patients based on the following indices (SVV < 13%, MAP > 75 mmHg, CI ≥ 2.0 L/min/m^2^, SVRI ≤ 3000 dynes s cm^−5^/m^2^). HR: heart rate; MAP: mean arterial pressure; CVP: central venous pressure; BIS: bispectral index score; GDFT: goal-directed fluid therapy; CI: cardiac index; SVRI: systemic vascular resistance index; SVV: stroke volume variation.

## Data Availability

Not applicable.

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
