# Peer review of "Analgesic Effect of Intrathecal Morphine Combined with Low-Dose Bupivacaine on Postoperative Analgesia after Liver Resection: A Randomized Controlled Study"

_jpm, 2022, doi:10.3390/jpm12020211_

Round 1

Reviewer 1 Report

Greetings

I read the manuscript with interest and find it unique. The hypothesis tested (beneficial effect of small dose bupivacaine to intrathecal morphine for additional analgesia after liver resection surgery) is interesting. Pain management after liver resection is challenging, thus, finding an optimal approach to intrathecal analgesia has practical values. However, I have a few comments to make

Abstract: Page 1, line 18- both groups were comparable to each 18 other… both groups is not clear. You can mention (M and M+B) in the bracket

Introduction: page 1, line 37-40. So far I know, post-resection coagulopathy is a major concern for continuing epidural analgesia.. and I believe, it should be included.

Methodology: Sample size calculation is missing

Why you have included both open and lap surgeries. Both have different pain intensities. Have you compared them? Were they distributed equally?

Concern about double-blinding: Control group received a percutaneous injection and the intervention group received SAB. There is an obvious difference in CSF flow. How blinding was possible? Further, in the immediate post-op, M+B will have the motor and sensory effects, which will reveal the group allocation indirectly? How blinding was possible here?

Was any patient on chronic analgesic use preoperatively, especially opioids? Were they excluded?

Results: CONSORT flow diagram can be added

Sedation has an impact on pain reporting. Was sedation monitored and compared? Please provide data.

Discussion: You have noted increased pruritus in the M+B group as compared to the M group while both groups received equal intrathecal morphine. Please discuss the possible reason

Analgesia consumption was higher in the M+B group as compared to the M group despite having additional bupivacaine. Please discuss the possible reason.

In the background- you have mentioned bupivacaine was added as morphine is hydrophilic and takes nearly 6 hours. As your surgery was approximately 3-4 hours, it is prudent to present fentanyl consumption within 3 hours postop.

Tables: It is prudent to present and compare fentanyl consumed per kg in the postop as well to find the real difference.

Overall comment: The manuscript is well written. Full form of IV (intravenous should be mentioned). Language and grammar are good. The methodology has some limitations. Still, I personally believe, authors should be given one chance to clarify/provide additional data, if available.

Best of luck

Author Response

We would like to thank you for your time and effort in reviewing our manuscript and providing your valuable comments, which have greatly helped us in revising our manuscript. We have carefully read your comments and have revised the manuscript. Our detailed response to each comment is written below.

Point 1: Abstract: Page 1, line 18- both groups were comparable to each 18 other… both groups is not clear. You can mention (M and M+B) in the bracket

Response 1: Thank you for your comment. We apologize for this mistake and I have made the edit in the text found in Page 1 line 18.

Point 2: Introduction: page 1, line 37-40. So far I know, post-resection coagulopathy is a major concern for continuing epidural analgesia.. and I believe, it should be included.

Response 2: Thank you for this comment. This is an important point. For patients undergoing liver resection, there is a possibility for disturbance in coagulation profile with highest risk for coagulation abnormalities appearing around the first day post surgery and normalizing around the fifth day after surgery. For these patients, there is high concern for increased risks of spinal hematoma and continuous placement of an epidural catheter can be debated against for liver resection patients for this reason.

I added the following in the text fount in Page 1 line 40-42.

Continuous placement of an epidural catheter after liver resection is a major concern because of post operation coagulopathy and increased risks of spinal hematoma.

Point 3: Methodology: Sample size calculation is missing

Response 3: Sample size calculation can be found in page 4, part 2.6 Power of study.

Point 4: Why you have included both open and lap surgeries. Both have different pain intensities. Have you compared them? Were they distributed equally?

Response 4: Thank you for this comment. We agree with this point. However, our study included liver resection which did not discriminate against type of surgery open or laparoscopic. The point that you make is a very interesting point and perhaps in a follow up study I think we should address this because there is increasing support for laparoscopic hepatic resection over open resection. In our table 1 Demographic data and intraoperational characteristics, we did compare operation type (open vs laparoscopy) amongst the three groups, which we did not find any significant difference.  Therefore, open and laparoscopic surgeries are equally distributed among three groups.

Point 5: Concern about double-blinding: Control group received a percutaneous injection and the intervention group received SAB. There is an obvious difference in CSF flow. How blinding was possible? Further, in the immediate post-op, M+B will have the motor and sensory effects, which will reveal the group allocation indirectly? How blinding was possible here?

Response 5: This was an important concern for us as well and we made sure to take this into careful consideration. The intrathecal injection was performed by trained anaesthesiologist otherwise uninvolved in rest of the study. This point was made in the following:

Original text: Page 2, 2. Materials and Methods

Patient group assignment, drug preparation, and drug administration was provided by a trained anaesthesiologist.

Revised:

Patient group assignment, drug preparation, and drug administration was provided by a trained anaesthesiologist uninvolved in the rest of the study.

Also, in the PACU (post-operative anaesthesia care unit), another independent anaesthesiologist not involved in this study who was blinded to allocation to the group assessed immediate post-operative side effects and VAS scores without bias of group allocation.  

Original text: Page 4

At 30 and 60 minutes upon entering the PACU, an anaesthesiologist blind to the patients’ assignment assessed whether the patient presented any side effects of the intrathecal injection by assessing the patient's sensory and motor block levels and whether the patient has a tingling sensation.

If you think it is necessary to clarify this further, I will make further editions because I agree that this very important. Thank you very much for your insightful comments.

Point 5: Was any patient on chronic analgesic use preoperatively, especially opioids? Were they excluded?

Response 5: No patients were on chronic analgesic use prior to surgery nor opioid.

Point 6: Results: CONSORT flow diagram can be added.

Response 6. Thank you for this comment. I have added the following consort flow diagram as Fig 1 in p.5.

Reviewer 2 Report

There is a small mistake on line 13 in the abstract section. Instead of 400 mcg morphine, 400 mg of morphine is written (must be written  micrograms, not milligrams, should be correct this false )

Line 128:what is this nefcom? should be explained.

Line 129:The brand of the drug used, the country of mnufacture should be written in parentheses.

Line 353:In the discussion section, it would be good to include some more study results.

Line 361:In the conclusion, it should be emphasized that added bupivacaine does not provide additional benefit,and morphine alone provides very effective analgesia.

Author Response

Response to Reviewer 2 Comments

We would like to thank you for your time and effort in reviewing our manuscript and providing your valuable comments, which have greatly helped us in revising our manuscript. We have carefully read your comments and have revised the manuscript. Our detailed response to each comment is written below.

Point 1: There is a small mistake on line 13 in the abstract section. Instead of 400 mcg morphine, 400 mg of morphine is written (must be written  micrograms, not milligrams, should be correct this false )

Response 1: Thank you for comment. We apologize for our mistake. We made the edit per your comment. 400 mcg.

Point 2: Line 128:what is this nefcom? should be explained.

Response 2: Thank you for this comment. I revised the text in the following:

Nefopam hydrochloride

Point 3: Line 129:The brand of the drug used, the country of mnufacture should be written in parentheses.

Response 3: I have edited the text per your comment. Thank you.

(Acupan, Korea)

Point 4: Line 353:In the discussion section, it would be good to include some more study results.

Response 4: Thank you for your insightful comment. Studies involving intrathecal local anaesthetics for major abdominal surgery under general anaesthesia are limited in literature which is one of the reasons we were interested in this subject. Postoperative pain control for liver resection is very tricky at times and optimizing a multimodal analgesic regimen would be helpful to the surgical and anaesthesiology team. The studies we found especially Koning MV et al. were included in our discussion. We think our study results will help increase these limited results. Thank you for this comment.

Point 5: Line 361:In the conclusion, it should be emphasized that added bupivacaine does not provide additional benefit,and morphine alone provides very effective analgesia

Response 5: Thank you for your comment. This is a very important point that you make. We also agree that low dose bupivacaine was insufficient to provide additional analgesic benefits to intrathecal morphine and intrathecal morphine alone provided effective analgesia. Also we were tentative about higher doses of bupivacaine bringing unwanted side effects.

We modified the text as the following:

Intrathecal morphine alone can be effectively implemented in a multimodal analgesic ap-proach in reducing overall opioid consumption in the postoperative care of liver resection in the first 24 hours. Addition of 5mg bupivacaine was insufficient in providing analgesic benefits in combination with intrathecal morphine, but resulted in sensory and motor block and tingling sensation in the case of hepatectomy. Therefore, intrathecal analgesia may warrant additional study for the optimal regimen.

Round 2

Reviewer 1 Report

Greetings

I read your revised manuscript and responses to the comments made. I applaud your work and I believe the revision has improved the manuscript. While your explanation regarding maintenance of blinding - I still have doubt and I feel it is required to be addressed. In modular OT (like us) camera attached with the light remains focused on the procedure site and the patient and others can usually observe the procedure being performed. I hope, such a condition in your set-up did not affect the blinding. 

Thank you

Author Response

We would like to thank you again for reviewing our manuscript and helping us to improve our manuscript. We have carefully read your comment and revised the manuscript accordingly. Our detailed response is written below. Again, we are very grateful for this opportunity to revise our manuscript.

Point 1: I read your revised manuscript and responses to the comments made. I applaud your work and I believe the revision has improved the manuscript. While your explanation regarding maintenance of blinding - I still have doubt and I feel it is required to be addressed. In modular OT (like us) camera attached with the light remains focused on the procedure site and the patient and others can usually observe the procedure being performed. I hope, such a condition in your set-up did not affect the blinding. 

Response 1: Thank you for this insightful comment. The cameras from our OT of our setting are screened at three different sites in our OR, from which our researchers were all excluded from. During the time of this study, the cameras did not record and allow screenings for patients in our country so the patients did not have access watch the recordings and see the procedures being performed. After the anesthesiologist performed the spinal procedure, he or she assessed the initial dermatomal level and was excluded for the operation room. The attending anesthesiologist was excluded from the operating room until the spinal procedure and initial dermatomal assessment was complete.  Thereafter, the attending anesthesiologist initiated, who was blinded to group allocation initiated general anesthesia. I have added the following to the text to elaborate on our blinding methods.

Revised text: Page 2, 2. Materials and Methods

Patient group assignment, drug preparation, and drug administration was provided by a trained anaesthesiologist uninvolved in the rest of the study. To ensure blinding, the attending anesthesiologist was excluded from the operation room until the spinal procedure and initial dermatomal assessment was complete. Thereafter, the attending anesthesiologist who was blinded to group allocation initiated general anesthesia.